

# Inventory of dams in Germany

Gustavo Andrei Speckhann[1,2], Heidi Kreibich[1], Bruno Merz[1,2]

[1]Helmholtz Centre Potsdam, GFZ German Research Centre for Geosciences, Potsdam, 14473, Germany
[2]Institute for Environmental Sciences and Geography, University of Potsdam, Potsdam, 14469, Germany

*Correspondence to*: Gustavo Andrei Speckhann (gustavo.speckhann@gfz-potsdam.de)

**Abstract.**

Dams are an important element of water resources management. Data about dams are crucial for practitioners, scientists and policymakers for various purposes, such as seasonal forecasting of water availability or flood mitigation. However, detailed information on dams on the national level for Germany is so far not freely available. We present the most comprehensive,

open-access dam inventory for Germany (DIG) to date. We have collected and combined information on dams using books, state agency reports, engineering reports, and internet pages. We have applied a priority rule that ensures the highest level of reliability for the dam information. Our dam inventory comprises 530 dams in Germany with information on name, location, river, start year of construction and operation, crest length, dam height, lake area, lake volume, purpose, dam structure and building characteristics. We have used a global, satellite-based water surface raster to evaluate the location of the dams. A

significant part (63%) of dams has been built between 1950-2013. Our inventory shows that dams in Germany are mostly single-purpose (52%), 53% can be used for flood control, and 25% are involved in energy production. The inventory will be freely available through GFZ Data Services (https://doi.org/10.5880/GFZ.4.4.2020.005) after discussion. In the meantime the data can be downloaded using the following temporary link https://dataservices.gfz-potsdam.de/panmetaworks/review/09c134e13a7ede5d80d67f641d615698c05917fac25fc776240a110d2df96a66/

(Speckhann et al., 2020).

## 1 Introduction

Dams are an important element of water resources management. Their main function is to store water in order to balance variations in streamflow and in demand of water and energy (World Commission on Dams, 2000). Most of the daily activities rely, to some extent, on the existence of dams: natural hazard mitigation (flood and drought control), energy

production (hydropower plants), food production (irrigation), transport of goods (inland navigation), and the daily water used in our houses and industries (water supply) (Bertoni et al., 2019; Gernaat et al., 2017; Moran et al., 2018; Poff and Olden, 2017; Sordo-Ward et al., 2012, 2013; Tullos et al., 2009).

Dams are classified as multi-purpose dams (i.e. the same structure is used for more than one purpose like water supply and flood control) or single-purpose dams. Multi-purpose dams can address multiple conflicting demands at different times,

i.e. dams may have one purpose (e.g. water supply) during normal hydrological conditions and have another purpose during



high precipitation periods in order to serve as a buffer for flood attenuation (Quinn et al., 2019). The ability to perform adequately under different tasks place multi-purpose dams as a potential answer to multiple societal challenges (Zarfl et al., 2019).

Dam operation is a challenge, especially due to the complexities and the non-linearity of the operations which may be driven by anthropogenic and environmental influences (Gutenson et al., 2020). For instance, a drought following the 2018 Kerala flood in India was worsened because dams had been emptied in preparation for floods, but the dam's water release combined with intense precipitation downstream aggravated downstream the flood and its impacts (Lal et al., 2020). In addition to such complexities, dam operators are compelled to decide in the short term (Mediero et al., 2007). Simple operating rules may not be ideal for a proper representation of the flow downstream, and to assume normal hydroclimatic conditions while

considering a restricted number of operating objectives has its limitations (Giuliani et al., 2016). On the other hand, nonlinear operating policies are not trivial to understand so that many dam operators tend to apply simpler rules, even though they may not provide the best answer for multiple-purpose conflicts (Quinn et al., 2019).

According to the International Commission on Large Dams (ICOLD, 2019), the majority of dams throughout the world serve irrigation purposes, both in single-purpose and multi-purpose dams. Irrigation represents the largest consumptive

use of fresh water and is used at approximately 20% of the world´s agricultural land (World Commission on Dams, 2000). Hydropower dams, the second most common application, produce 72% of the global renewable energy (Gernaat et al., 2017). Many countries aim to expand the hydropower potential in the near future (Bertoni et al., 2019; Timpe and Kaplan, 2017; Zarfl et al., 2014), especially due to its advantage to produce energy only when the electrical system demands it (Zarfl et al., 2019). Pacific Asia, South America and Africa have a significant potential for the construction of new hydropower plants (Gernaat et

al., 2017; Timpe and Kaplan, 2017), while in Europe and North America, the current number of dams being built is smaller than the number of dams being removed (Moran et al., 2018).

The construction of dams can lead to negative socio-economic and environmental consequences (Best, 2019; Zarfl et al., 2019). The relocation of people and conflicts regarding water use at different catchment locations can have a long-term effect on communities. Dams, and especially cascades of dams, can be a threat to biodiversity and a driver of species extinction

(Winemiller et al., 2016), and are responsible for geochemical cycle modifications at the global level (Maavara et al., 2020). Dams can aggravate stress on rivers streamflow by altering the hydrological regime (Bond et al., 2008; Zarfl et al., 2019; Zhang et al., 2019), and by modifying sediment fluxes along the river (Maavara et al., 2020; Manh et al., 2015; Winemiller et al., 2016). For instance, at the Amazon river, one of the most explored areas for hydropower plants, a reduction of 20% of the suspended sediment was observed (Latrubesse et al., 2017).

Information and data about dams are crucial for dam operators, scientists and policy makers (Hecht et al., 2019; Tullos et al., 2009), for instance, for the assessment of hydropower plants construction (Bertoni et al., 2019; Gernaat et al., 2017; Moran et al., 2018; Winemiller et al., 2016), estimation of the hydrological footprint (Bakken et al., 2013, 2014; Popescu et al., 2020; Postel, 2000), seasonal forecasting of water availability and water levels in rivers for navigation (Zhang et al., 2019), flood and drought risk assessment and management (Di Baldassarre et al., 2017; Ehsani et al., 2017; Elmer et al., 2012; Metin



et al., 2018; Veldkamp et al., 2017), and assessment of biotic disturbances (Latrubesse et al., 2017; Maavara et al., 2020; Sabo et al., 2017). The consideration of dams on hydrological models improves the model performance downstream (Gutenson et al., 2020), but the lack of information on dams may jeopardize its modelling applications, specially at a national level (Zhao et al., 2020).

Important information about dams are the location of the dam, the lake area, and volume. Often it is also important
to know when a dam was constructed or went into operation (and out of operation). The purpose of a dam is commonly known, but dam operation rules are normally not available at dam databases. Dam operations are normally complex and influenced by environmental and anthropogenic factors (Gutenson et al., 2020). Many studies have explored the effects of dam operation rules, however, operation rules are not easily available, especially for private hydropower plants (Bertoni et al., 2019; Giuliani et al., 2016; Mediero et al., 2007; Pan et al., 2014; Quinn et al., 2019).

In 1972, the congress of the United States of America authorized the first nationwide survey on dams, which was conducted by the U.S. Army Corps of Engineers (US Army Corps of Engineers, 1975). This was, to our knowledge, the first dam inventory conducted at a national level. Initially, the motivation was to collect data on the dam hazard level (i.e. likely life loss in case of dam failure), height, and storage capacity. This initiative continues until today, and the database has currently information on more than 90,000 dams.

The Global Reservoir and Dams database - GRanD (Lehner et al., 2011) was the first database to provide global information about dams with a detailed level of information concerning the location (municipality, country), dam height, surface area, purpose, dam crest, and elevation. The first version was made publicly available in 2011 with 6,862 dams, and an updated version was released in 2019 containing more than 7,000 dams. Recently, the GlObal geOreferenced Databased of Dams – GOODD, (Mulligan et al., 2020) was published, containing more than 38,000 dams (and their catchments) across the
whole globe. For the German territory, the GRanD database contains 60 dams, while the GOODD database contains 142 dams. As an outcome from the Sustainable Development Goal 6.6.1 the UN has freely provided Geotiffs files for all administrative boundaries and basins on water surface presence between the period of 2000-2018 at a resolution of 30 meters. The products available are the result from a joint cooperation between the European Commission's Joint Research Center in partnership with Google Earth.

The objective of this work is to provide, as comprehensive as possible, open-source information of dams in Germany. We have compiled data from multiple sources, such as federal agency reports or scientific publications. Our main source was the German Commission on Dams ([www.talsperrenkomitee.de](http://www.talsperrenkomitee.de)), especially their book published in 2013 which contains information on 340 dams (Deutsche TalsperrenKomitee e. V., 2013). We collected information about 530 dams in Germany and individually identified their location using satellite images from Google Earth. The database was evaluated via a
comparison with the available dam databases GRanD (Lehner et al., 2011) and GOODD (Mulligan et al., 2020), and the Sustainable Development Goal 6 product (Pekel et al., 2016). Our database contains significantly more dams than the compilation of the German Commission on Dams and combines different sources of information to provide detailed information, that are herewith made freely available.



## 2 Material and Methods

**2.1 Data compilation procedure**

There are no agreed standards regarding the collection of dam data. At a global level, there are only two open-access data publications on dams (Lehner et al., 2011; Mulligan et al., 2020). In order to create a national dam inventory for Germany (DIG), we have followed six steps (Figure 1). The first step was to search for information about dams across a variety of sources: books, journals, federal agencies reports and web-pages. We searched for any reference in English or German that

may lead to some information about dams in Germany. We have searched for any reference to "dams" and/or "Germany" at libraries and internet search engines. This yielded a considerable number of possible sources. The most comprehensive source was the book "Talsperren in Deutschland" (Deutsche TalsperrenKomitee e. V., 2013) from where a significant part of the data in this inventory was obtained.

In the second step, the dams were geolocated using satellite images. Only dams that could be identified visually were

included in our database. The identification was carried out using Google Earth, based on the combination of the name of the dam and the city where it was located. For those cases where a dam could not be identified, additional information on the dam location was used to increase the chance to find it (e.g. name of the river, neighborhood, image).

All identified dams were included in the inventory with their name and coordinates (step 3). The dam coordinates were a condition *sine qua non* for the inclusion of the dam in the inventory. We have not established a threshold for the

inclusion, e.g. based on dam height or lake volume, but we have added only dams for which we could provide reliable information. As such, we kept the database as useful as possible for different purposes.

In the fourth step different sources of information were combined to add the missing data to each dam entry. In many cases the information agreed between different sources. In the case of conflicting information, the following hierarchy of information sources was applied taking into consideration the perceived reliability of the sources: 1) book of dams in Germany

(Deutsche TalsperrenKomitee e. V., 2013); 2) state agencies reports (e.g. Landesanstalt für Umwelt Baden-Württemberg); 3) other dam databases (Lehner et al., 2011; Mulligan et al., 2020); 4) engineering reports; 5) web-pages. In order to have as much information as possible for each dam, the search was an iterative process, where the priority rule was applied during every round.

The fifth step was the evaluation of the dam's location and it was conducted as a validation. We used the location and

persistence of surface water obtained from the analysis of Landsat satellite images (Pekel et al., 2016) as a proxy for the presence of the dam's location. Using a GIS software, we compared the location of the dam with the water surface occurrence by looking at the intersection of the dam buffer and the water surface presence. We used the latest available grid information on surface water, downloaded from the United Nation Program on Sustainable Development Goal 6.6 (https://www.sdg661.app/downloads), to indicate the presence of water and therefore confirm that a water surface was spotted

at the dam's location.





The last step was to make the inventory publicly available at the GeoForschungsZentrum Data Service (http://dataservices.gfz-potsdam.de/portal/), where a shapefile and tab-delimited file containing all information can be downloaded.

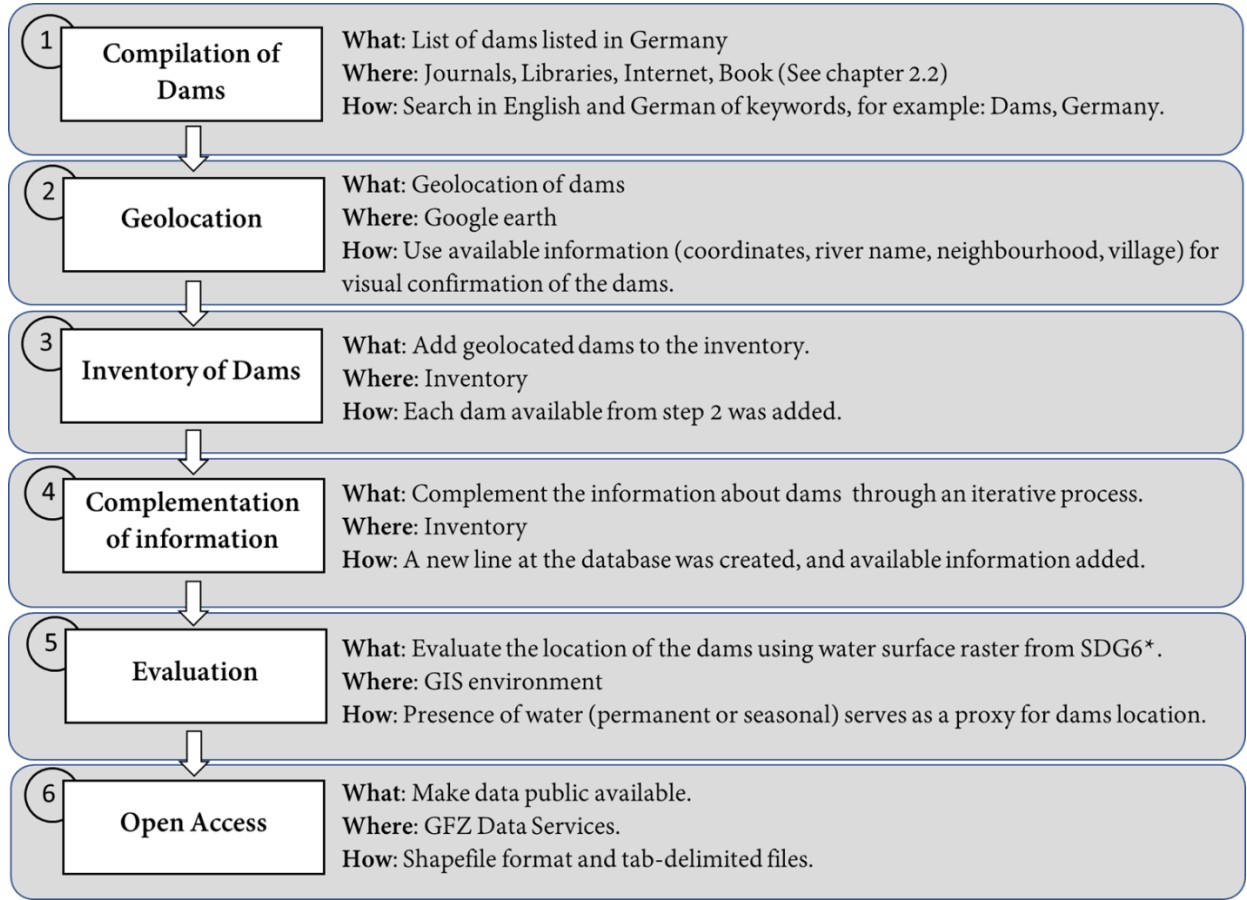

135        **Figure 1** - Procedure for the compilation of the dam inventory. *Sustainable Development Goal 6 app application. (**https://www.sdg661.app/data-products/data-downloads**)

## 2.2 Data sources

The information available was classified into five main sources as described under step 4 and listed in detail in Table 1. The book of dams in Germany (Deutsche TalsperrenKomitee e. V., 2013) is the most detailed information source providing

technical information for 350 dams in Germany. State agency reports with dam information are available from all states in Germany, except for Saarland, Rhineland-Palatinate, Schleswig-Holstein, and Mecklenburg-Western Pomerania. A possible explanation for that fact is the smaller number of dams in those states. On the other hand, Baden-Württemberg, North Rhine-Westphalia and Bavaria provide dam information on the websites of their environmental agencies. The two global databases tend to focus on big dams using automated algorithms for reservoir identification based on topography. The GRanD database



by Lehner et al. (2011) is the second most complete set of dam information for Germany, containing information of geolocation, dam height, purpose and lake area for 60 dams in Germany. The GOODD database (Mulligan et al., 2020) covers 142 dams in Germany but contains no information apart from the geolocation and watershed size. The global databases are the only information that can be used directly in a GIS environment and provide coordinates for each dam. The information apart from the global databases are only available as document pages that cannot be exported to a user-friendly format; the use of

this information requires to copy the information manually for each dam. Some of the information from state agencies files (tables, texts) are protected and do not allow any selection. Engineering reports are normally available for more recent dams or for dams that have been under restoration or retrofitting, e.g. increase in dam capacity. Web-pages are also a source of dam information. However, verifiable references or links are often missing, and in some occasions differences in naming the dams occurred. Hence, we decided to avoid web-pages as the main information source of a dam. Their information was used as

complementary data and was combined with other sources. Table 1 presents the most relevant sources of information used to compile the DIG.

**Table 1** The most relevant sources of information used to compile the DIG.

| Source of information | Information contained | Number of dams in Germany | Format availability | Region |
|---|---|---|---|---|
| Deutsches TalsperrenKomitee e.V. | Name of the dam, start of operation, state, river name, dam height, lake area, lake volume, construction type, building characteristics | 348 | Book (table) | Germany |
| Bayerisches Landesamt für Umwelt | name of the dam, start of operation, state, river name, crest length, dam height, lake area, lake volume, construction type, purpose, building characteristics, coordinates | 24 | Internet page | Bavaria |
| Hessisches Ministerium für Umwelt, ländlichen Raum und Verbraucherschutz | Name of the dam, start of operation, state, river name, lake area, lake volume | 45 | Table (pdf) | Hessen |
| Landesamt für Natur, Umwelt und Verbraucherschutz Nordrhein-Westfalen | Name of the dam, start of operation, state, crest length, lake volume, operator/responsible | 44 | Table (pdf) | North Rhine-Westphalia |
| Landesamt für Umwelt Brandenburg | Name of the dam, start of operation, state, river name, dam height, lake volume, purpose, building characteristics | 2 | Internet page | Brandenburg |
| Landesanstalt für Umwelt Baden-Württemberg | Name of the dam, start of operation, state, river name, crest length, dam height, lake area, lake volume, construction type, purpose, building characteristics | 80 | Table (pdf) | Baden-Württemberg |



| Niedersächsischer Landesbetrieb für Wasserwirtschaft, Küsten- und Naturschutz | Name of the dam, state, river name, lake area, lake volume, purpose | 12 | Internet page | Lower Saxony |
|---|---|---|---|---|
| Sächsisches Landesamt für Umwelt und Geologie Abteilung Wasser | Name of the dam, start of operation, state, river name, crest length, dam height, lake area, lake volume, construction type, purpose, building characteristics, operator/responsible | 162 | Table (pdf) | Saxony |
| Talsperrenbetrieb Sachsen-Anhalt | Name of the dam, start of operation, state, river name, dam height, lake volume, purpose, | 32 | Internet page | Saxony-Anhalt |
| Thüringer Fernwasserversorgung | Name of the dam, start of operation, state, river name, crest length, dam height, lake area, lake volume, construction type, purpose, building characteristics, | 7 | Internet page | Thuringia |
| GOODD database (Mulligan et., 2020) | Geolocation, watershed, size | 142 | Paper (shapefile available) | Global |
| GRanD database (Lehner et al., 2011) | Name of the dam, start of operation, state, river name, crest length, dam height, lake area, lake volume, construction type, purpose, geolocation, watershed size | 60 | Paper (shapefile available) | Global |

# 3 Results

## 3.1 Data contents

Our dam inventory contains: name of the dam, date when the construction and operation started, state, river, length of the crest, height of the wall, area of the lake at full capacity, volume at full capacity, dam type, building characteristics, and the geolocation (Table 2). Information on the type, building characteristics and purpose is given as acronyms; a table containing these acronyms is provided in the folder for download. We have compiled the most comprehensive dam inventory for Germany, however, we believe that there are more dams in Germany that for many reasons were not identified via our

procedure. Hence, there is room for expanding our inventory.

**Table 2- Data present in the dam inventory with units.**

| Variable | Unit | Description |
|---|---|---|
| Name | - | Name of the dam. |
| Start date of construction | - | Year in which dam construction started (YYYY). |
| Date of operation | - | Year in which the construction was finished (YYYY). |
| State | - | Represented by two letters (e.g. BY - for Bavaria). |



| River | - | Name of the river where the dam is located or close to. |
|---|---|---|
| Crest length | m | Length of the dam crest. |
| Dam height | m | Dam height from base to top. |
| Lake area | km² | Lake area at the full capacity. |
| Lake volume | Mio.m³ | Lake volume at the full capacity. |
| Construction type | - | Type of structure (e.g. barrage, pond). |
| Purpose | - | Dam function(s) (e.g. flood control, energy production). |
| Building characteristics | - | Structural dam characteristics (e.g. embankment dam). |
| Location | - | Geometric coordinates using WGS84. |

### 3.2. Data characteristics

Our dam inventory comprises information about 530 dams which are in operation in Germany (Figure 2). Saxony is
the state with the highest number of dams in our inventory with a total of 190 dams. North Rhein-Westphalia is the second one
with 83 dams followed by Baden-Württemberg with 79. For the city-states of Berlin, Bremen and Hamburg no dams were
found. Brandenburg (2), Mecklenburg-Western Pomerania (2), Rhineland-Palatinate (4), Saarland (3) and Schleswig-Holstein
(1) have a small number of dams (Figure 2). These states are located at a lower elevation. The majority of dams in the remaining
states Thuringia (43), Bavaria (44), Saxony-Anhalt (32), Hesse (28), and Lower Saxony (19) are located at an altitude between
100 and 500 meters. The number of dams is a function of the data availability i.e. states may have more dams on their territory
but the information was not publicly available or the dam not detected by satellite image algorithms. We do not claim to have
compiled an exhaustive list of all dams in Germany.

Our inventory contains 276 single-purpose dams and 254 multi-purpose dams. Flood control (HWS) is the most
common purpose (Figure 3a) often in combination with other purposes. Only 36% out of 284 dams with flood control function
are used exclusively for flood control. Recreation (NEG) is the second most common purpose 179 dams, whereas the majority
(86%) are multi-purpose dams. Energy production (E) is also widespread with 131 dams. 23 dams are used for nature protection
(NSG). In order to assure a minimum level of navigability at waterways, especially during summertime, 62 dams are used to
regulate the river water level for transport reasons (NWA). Our inventory contains 130 dams for water supply (TWv) which
are spread across the country. Industrial and agriculture water supply (BWv) are an important component for food (e.g.
irrigation) and energy production (e.g. cooling of nuclear power plants) and other industry-related activities with 91 dams.



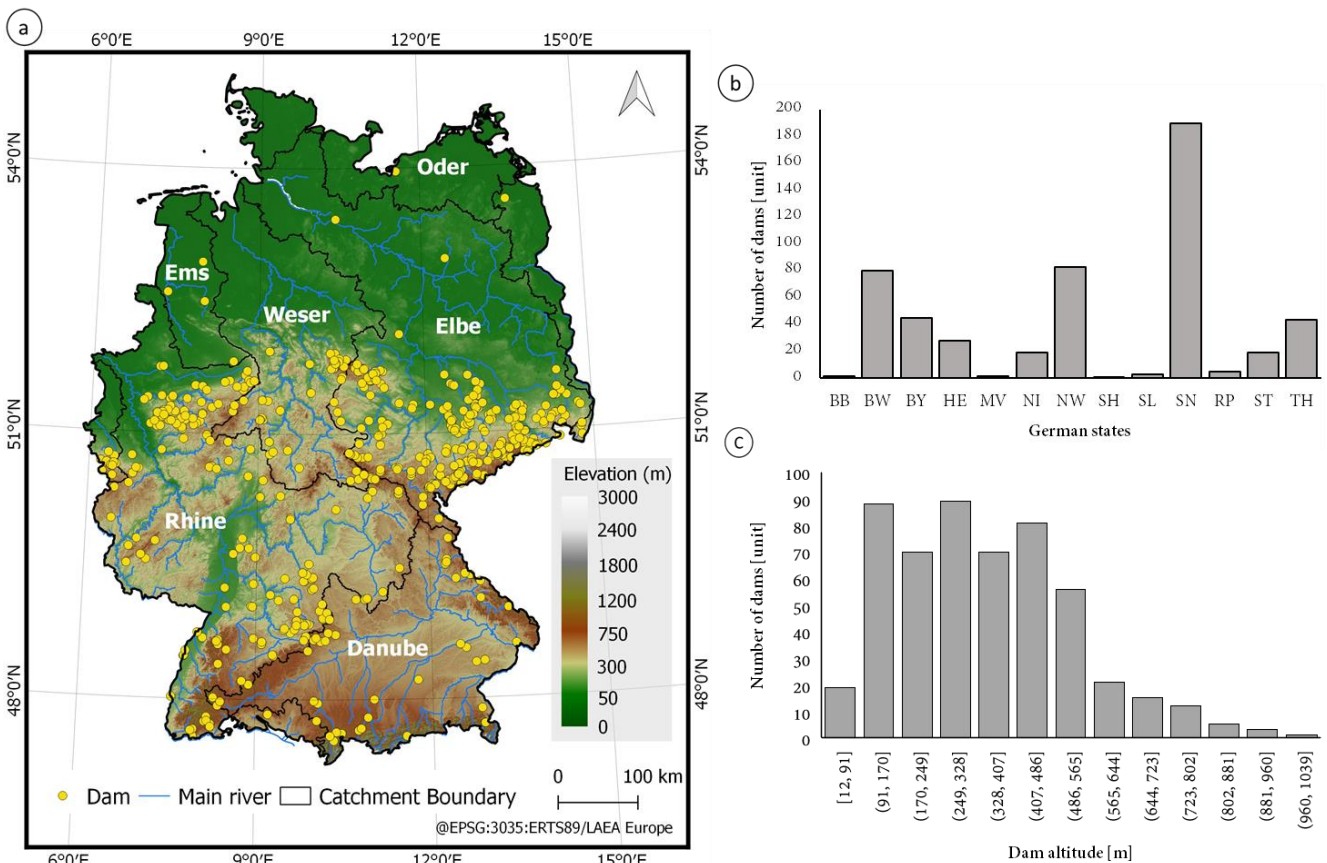

**Figure 2 – Dam locations across Germany (a), number of dams in the German states (b), and dam altitude (c). State acronyms are: BB (Brandenburg), BW (Baden-Württemberg, BY (Bavaria), HE (Hesse), MV (Mecklenburg-Western Pomerania), NI (Lower Saxony), NW (North Rhein-Westphalia), SH (Schleswig-Holstein), SL (Saarland), SN (Saxony), RP (Rhineland-Palatinate), ST (Saxony-Anhalt), TH (Thuringia).**


The building characteristics are dominated by three types: embankment dams, gravity dams, and rockfill dams (Figure 3b). Embankment dams (EDD) are by far the most frequent type (289 dams, 54%), followed by gravity dams (97 dams, 18%) and rockfill dams (75 dams, 14%). No relation was observed between the building characteristics and the states.



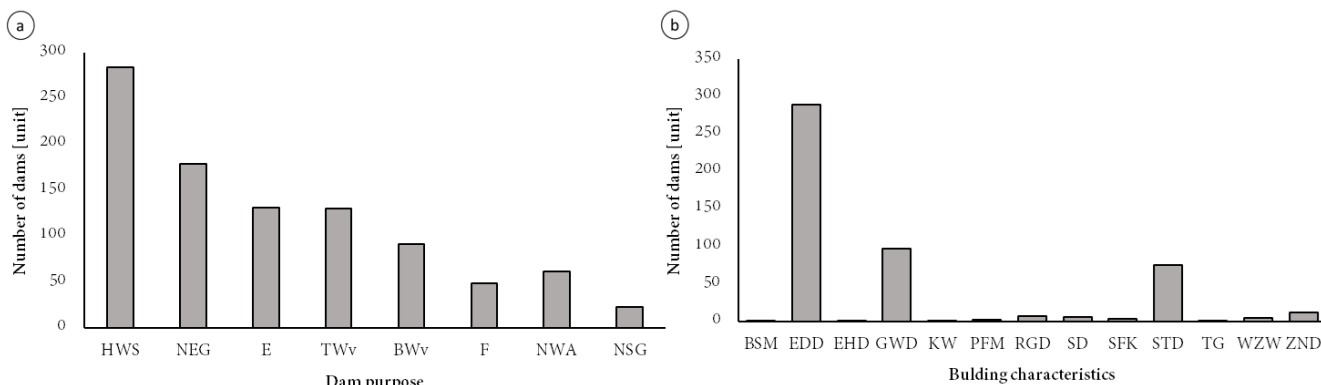

**Figure 3 - Dam purpose (a) and building characteristics (b). Purposes are abbreviated as: BWv (industrial agricultural water supply), E (energy production), F (fishing), HWS (Flood control), NEG (recreational), NSG (nature protection), NWA (transport) and TWv (water supply). Building characteristics are abbreviated as: BSM (arch dam), EDD (embankment dam), EHD (homogenous rockfill dam), GWD (gravity dam), KW (flap weir), PFM (buttress dam), RGD (ring dam), SD (fill dam), SFK (segment with fish belly flap), STD (rockfill dam), TG (residual lake associated with mining), WZW (rolling weir), ZND (zone dam).**

After the second world war, many countries initiated the construction of dams with a particularly strong increase in Europe and US after 1950 and 2000 (Moran et al., 2018). This trend is also seen in Germany (Figure 4). Between 1950 and 2000 more than 300 dams were constructed, but only 20 dams after 2000 were built. The reasons for this reduction are twofold: 1) the decrease in available locations where dams can be constructed, and 2) the chance of structural failure associated to the dam existence and operation.

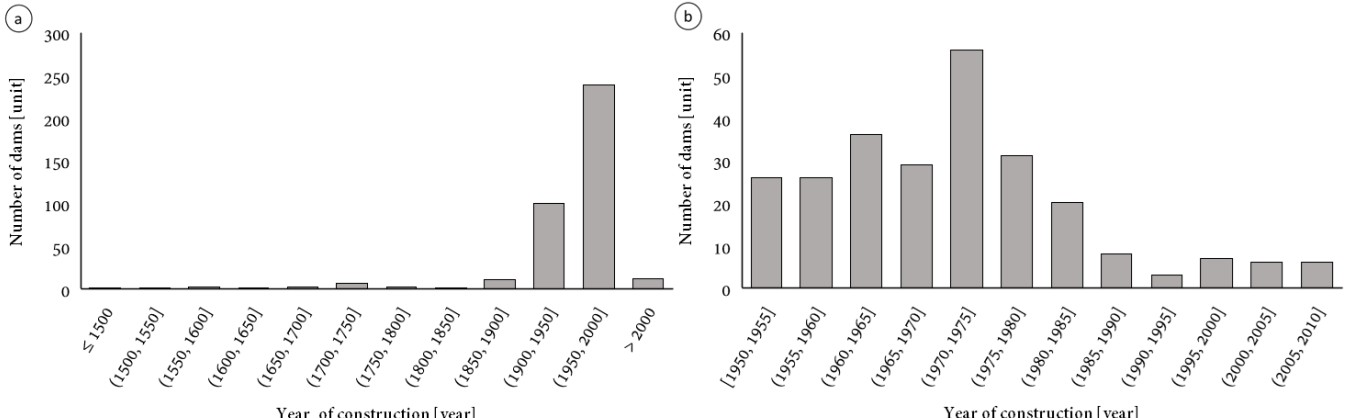

**Figure 4 – Construction year of the dams in Germany (a) and a subplot for the period between 1950-2015 (b).**

Dam height is one of the criteria used to classify dams. The International Commission on Large Dams (ICOLD, 2011) defines dams with heights above 15 meters as large dams. 266 dams (50%) in our inventory are considered as large dams (Figure 5b). We could not observe a relation between dam height and the states.





There are 231 dams (43%) with a lake volume smaller than 1 Mio m³. Dams tend to be decentralized, i.e. there are rather several smaller dams which may act together, instead of one massive dam within a region. The DIG has 11 dams with lake volume higher than 100 Mio m³ and the majority of them were constructed for energy production and flood control.

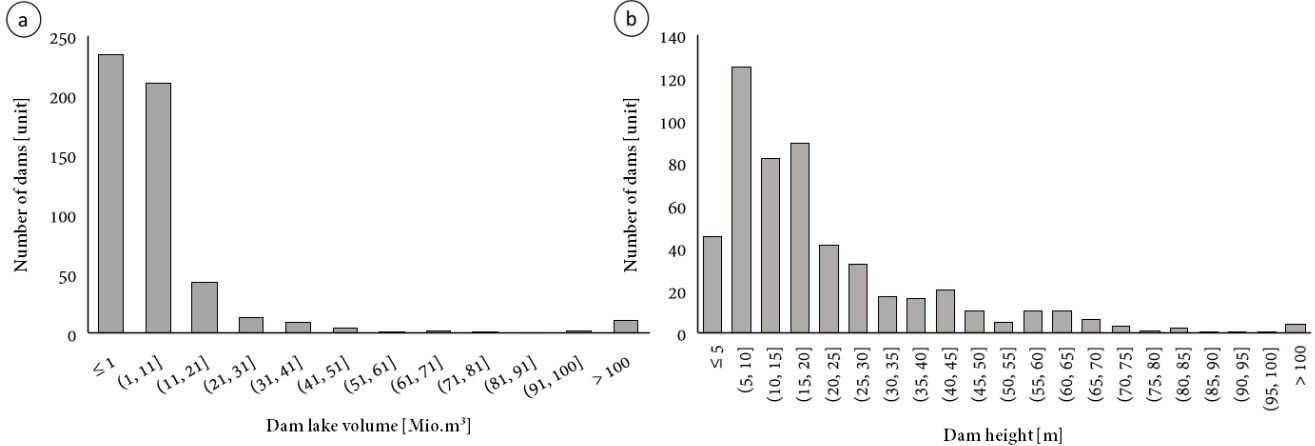


**Figure 5 – Lake volume (a) and dam height (b).**

### 3.3 Evaluation of location of dams

Dam location is central in our inventory, so we evaluated it using global data. We used water surface raster information for the year 2018 (Pekel et al., 2016) as a proxy for the location of active dams. This raster is derived from Landsat images,
however, they may not be able to correctly identify dams with a lake area smaller than 0.3 km² and lake width smaller than 30 meters, respectively (Lehner et al., 2011; Mulligan et al., 2020). For each dam in the inventory, we created a buffer of 200 meters around a point and used that polygon as a mask to extract the values from the 30-meter resolution water surface raster. The raster contains 3 possible states: 'permanent water', 'seasonal water', and 'no water'. Each dam location was investigated for the presence of permanent water or seasonal water for the year 2018.

From the 530 dams at the inventory, 81 dams did not exhibit the presence of water (permanent or seasonal). 63 of those dams are classified as single-purpose dam for flood control. The majority of the remaining dams (14 out of 18 dams) are used for flood control together with other purposes and have a lake area smaller than the threshold for lake identification (0.3 km²). As single-purpose flood control dams are only used in flood situations, it is expected that most of them are not identified by a satellite-derived water surface product for a given year, as they may not have experienced a flood situation in this year or
due to cloud interference during floods. Thus, we believe that our evaluation of the dam location is sufficient for compiling the dam inventory.

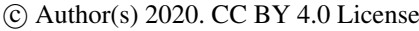

## 4 Data availability

During the discussion period data can be freely downloaded via GFZ Data Services at https://dataservices.gfz-potsdam.de/panmetaworks/review/09c134e13a7ede5d80d67f641d615698c05917fac25fc776240a110d2df96a66/. After the
review process, the data will be accessible via https://doi.org/10.5880/GFZ.4.4.2020.005 (Speckhann et al., 2020).

## Conclusions

We compile an open-access inventory of dams at the national level in Germany. This information can be used by scientists, practitioners and policy makers for a range of purposes, such as flood and drought risk assessment and management or assessment of biotic disturbances. We have built the most comprehensive inventory of dams in Germany by providing name
of the dam, name of the river, technical specifications (dam height, crest length, building characteristics, purpose and type) as well as the dam geolocation. With information about 530 dams, our inventory comprises almost 1.5 times the number of dams than the second most comprehensive source, i.e. the book of the German commission on large dams. The inventory consists of a shapefile format which enables the easy use of this information at GIS environments and modelling applications and also a tab-delimited file that can be easily imported for other uses. As states in Germany advance the digitalization together with
public administrative transparency, we believe that there will be opportunities for the DIG inventory to be updated and extended in the future, both in respect to the number of dams as well as the amount of information.

## Author contribution

GS and HK and BM designed the study and GS carried it out. GS prepared the manuscript with valuable contributions from all co-authors.

## Competing interests

The authors declare that they have no conflict of interest.

## Acknowledgements

The first author would like to thank the German Academic Exchange Service (DAAD) for the financial support during the research.

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
