# Peer review of "Inventory of dams in Germany"

_Earth System Science Data, 2020_

## Referee Comment (RC1) · Anonymous Referee #1 · 23 Nov 2020

This manuscript describes the review of dams in Germany. As a non-research paper it can mostly receive technical comments. However, I think that the manuscript overall is very comprehensive, well written and includes the necessary information for the reader to understand how and to what extent the review was conducted by the authors. It really seems an extensive review. The manuscript begins with an informative introduction on dams, especially on their operations and their role for the environment and economy, as well as on the importance for scientists and researchers to have access to information on dams. The information currently available is also summarized through references to other inventories both in Germany and worldwide, with special references to the databases GRanND and GOODD. Next, the manuscript gives all the different sources used in this Germany dams inventory and in the results presentation it categorizes the dams appropriately (according to the purpose, building characteristics, lake volume etc) providing useful and informative graphs. Overall, I think that both the procedure of

the dams inventory and the description of the final output (database along its contents) are fine. Therefore, I can only contribute with only two minor/technical comments: 1) Lines 35-37. I cannot easily follow the meaning of this sentence. Can this become clearer? 2) Lines 58-59: A 20% sediment (including suspended material) reduction seems too low I think. Other references could be found reporting much higher numbers (not only for Amazon).

———————————————————

---

## Referee Comment (RC2) · Benjamin Fersch (Referee) · 5 Jan 2021

The authors present a dataset of a nationwide inventory of dams for Germany. In addition to location information and operations startup date, additional important properties are listed.

The dataset is a valuable compilation that unifies information that is typically split due to the federal state structure of Germany and that is often not available in a usable digital format.

I have the following comments:

1. The title refers to dams in Germany, but the introduction focuses mainly on dams in other parts of the world and some global inventories. I think that a paragraph describing the situation (history of water management, federal structure) in Ger-

many would be interesting to the readers / users.

2. Section 3.2 lists the purposes of the dams in the inventory, but it's not clear if structures without considerable manageable reservoir volume (e.g., hydraulic power stations, channels, or locks) are generally considered, or excluded. These kind of structures usually don't alter the discharge too much. Some dams along the Lech river were considered, others, e.g., for the Danube were excluded.

3. Although there were many different sources used to fill the database, the very easy to exploit database of Openstreetmap (which exhibits a very good data quality for Germany) was apparently not used here. Using a simple query on http://overpass-turbo.eu/

```
[out:json][timeout:25];
// gather results
(
  // query part for: ``dam''
  node["waterway"="dam"]({{bbox}});
  way["waterway"="dam"]({{bbox}});
);
// print results
out body;
>;
out skel qt;
```

one can easily find many more reservoirs, that are of equal size / volume as those listed in the inventory. E.g., Herrenteich, Reinfeld (Holstein), Padersee, Paderborn, more than 50 reservoirs around Clausthal whereof 3 are listed in your inventory, Blaibacher See (Schwarzer Regen, near Viechtach) just downstream of the listed Hoellenstein Talsperre, or Baerensee and Bachtelsee (Werach) close

to Kaufbeuren. So in case you have no capacities to rework your inventory based on the OSM data source you may want to provide this hint in your outlook as a starting point for further investigations, also outside of Germany.

4. The dataset is restricted to the political boundaries of Germany. It would be nice if the extent was "hydrological" Germany including the whole catchments of Elbe, Rhine and Danube, but that would include Czechia, France, Austria and Switzerland. For hydrological modeling this would be desirable but I admit that this is out of scope of this work.
* * *

---

## Author Comment (AC1) · 20 Jan 2021

Authors response to Referee #1

Dear referee, we would like to thank you for the time and effort put into reviewing the manuscript. This response carefully addresses all the comments (our response is marked with an R:).

Best regards,

Gustavo Andrei Speckhann on behalf of all co-authors

Referee #1 This manuscript describes the review of dams in Germany. As a non-research paper it can mostly receive technical comments. However, I think that the manuscript overall is very comprehensive, well written and includes the necessary information for the reader to understand how and to what extent the review was con-

ducted by the authors. It really seems an extensive review. The manuscript begins with an informative introduction on dams, especially on their operations and their role for the environment and economy, as well as on the importance for scientists and researchers to have access to information on dams. The information currently available is also summarized through references to other inventories both in Germany and worldwide, with special references to the databases GRanND and GOODD. Next, the manuscript gives all the different sources used in this Germany dams inventory and in the results presentation it categorizes the dams appropriately (according to the purpose, building characteristics, lake volume etc) providing useful and informative graphs. Overall, I think that both the procedure of the dams inventory and the description of the final output (database along its contents) are fine.

R: Thank you very much for this positive feedback!

Therefore, I can only contribute with only two minor/technical comments: 1) Lines 35-37. I cannot easily follow the meaning of this sentence. Can this become clearer?

R: We have re-written this sentence as follows, to make it clearer: "For instance, a flood event was aggravated due to the opening of floodgates of upstream reservoirs in combination with extreme precipitation in 2018 in the Kerala region in India (Lal et al., 2020)."

2) Lines 58-59: A 20% sediment (including suspended material) reduction seems too low I think. Other references could be found reporting much higher numbers (not only for Amazon).

R: We have included the following additional information and references: "For instance, in Brazil it is estimated, that both the Paraná basin as well as the Tocantins basin, have a bed load trap of approximately 80% (Latrubesse et al., 2005). At the Amazon river, one of the most explored areas for hydropower plants, a reduction of 20% of the suspended sediment was observed (Latrubesse et al., 2017). Global estimations indicate that the majority of large dams have a theoretical sediment trapping efficiency

higher than 50% (Vörösmarty et al., 2003)."

Latrubesse, E. M., Stevaux, J. C. and Sinha, R.: Tropical rivers, Geomorphology, 70(3-4 SPEC. ISS.), 187–206, doi:10.1016/j.geomorph.2005.02.005, 2005.

Vörösmarty, C. J., Meybeck, M., Fekete, B., Sharma, K., Green, P. and Syvitski, J. P. M.: Anthropogenic sediment retention: Major global impact from registered river impoundments, Glob. Planet. Change, 39(1–2), 169–190, doi:10.1016/S0921-8181(03)00023-7, 2003.

Authors response to Dr. Benjamin Fersch #2

Dear Dr. Benjamin Fersch, we would like to thank you for the time and effort put into reviewing the manuscript. This response carefully addresses all the comments (our response is marked with an R:).

Best regards,

Gustavo Andrei Speckhann on behalf of all co-authors

The authors present a dataset of a nationwide inventory of dams for Germany. In addition to location information and operations startup date, additional important properties are listed. The dataset is a valuable compilation that unifies information that is typically split due to the federal state structure of Germany and that is often not available in a usable digital format. I have the following comments:

R: Thank you very much for this positive feedback!

The title refers to dams in Germany, but the introduction focuses mainly on dams in other parts of the world and some global inventories. I think that a paragraph describing the situation (history of water management, federal structure) in Germany would be interesting to the readers/users.

R: Thank you very much for pointing this. We have included the following additional sentence at the introduction of the paper: "In Germany, the first dams, i.e. fish and mill

ponds with a small reservoir volume, are documented from the end of the 8th century (Bettzieche 2010 ). The construction of dams gained importance when energy was needed for mining in the Harz and Ore mountains in the mid-16th century. Modern dam construction began with the Eschbach Dam (1891) and was driven by the goal of the industry in the low mountain range to remain competitive with the newly emerging industry in the Ruhr area (Bettzieche 2010). Later, other dam purposes such as flood control, recreation, and water supply became much more important. Most of the dams were built in Germany between 1950 and 2000 (Deutsches TalsperrenKomitee e. V., 2013). "

Section 3.2 lists the purposes of the dams in the inventory, but it's not clear if structures without considerable manageable reservoir volume (e.g., hydraulic power stations, channels, or locks) are generally considered, or excluded. These kinds of structures usually don't alter the discharge too much. Some dams along the Lech river were considered, others, e.g., for the Danube were excluded.

R: We have aimed to include as many dams as possible, without a threshold of volume. However, dams were only included in our dam inventory if information about most characteristics were available and verifiable. Although there were many different sources used to fill the database, the very easy to exploit database of Openstreetmap (which exhibits a very good data quality for Germany) was apparently not used here. Using a simple query on http://overpass-turbo.eu/

```
[out:json][timeout:25];
// gather results
(
// query part for:"dam"
node["waterway"="dam"]({{bbox}});
way["waterway"="dam"]({{bbox}});
```

);

// print results

out body;

>;

out skel qt;

one can easily find many more reservoirs, that are of equal size / volume as those listed in the inventory. E.g., Herrenteich, Reinfeld (Holstein), Padersee, Paderborn, more than 50 reservoirs around Clausthal whereof 3 are listed in your inventory, Blaibacher See (Schwarzer Regen, near Viechtach) just downstream of the listed Hoellenstein Talsperre, or Baerensee and Bachtelsee (Werach) close to Kaufbeuren. So in case you have no capacities to rework your inventory based on the OSM data source you may want to provide this hint in your outlook as a starting point for further investigations, also outside of Germany.

R: We appreciate very much the recommendation as well as the hint to the OSM query. We believe the OSM can provide valuable information about the location of the dams in Germany (also in Europe). However, a central aspect of our work was the compilation of highly reliable and detailed information about dams, not only their location. The information provided via OSM contributes mainly the location of the dam, but provides no (or little) details about the technical dam characteristics (e.g. crest length, dam height, start of operation). When additional information is provided, there is almost no reference to where the information was obtained and how reliable it is. We have included the following additional information to the manuscript: "The OpenStreetMap (OSM) is also a freely available online tool that contains information about dams globally. Through the www.overpass-turbo.eu, a web-based tool for filtering OSM, one can easily identify the location of dams."

The dataset is restricted to the political boundaries of Germany. It would be nice if

the extent was "hydrological" Germany including the whole catchments of Elbe, Rhine and Danube, but that would include Czechia, France, Austria and Switzerland. For hydrological modeling this would be desirable but I admit that this is out of scope of this work.

R: We have focused on dam information for Germany, so that compiling dam information in neighbouring countries is out of scope of our work. Still, we agree that a comprehensive database with dam information for all German catchments, including areas outside of Germany is a desired goal. Based on this comment, we have included the following additional information into the conclusions: "Further extensions of the inventory may also include dams in upstream regions of the Elbe, Rhine and Danube catchments outside Germany, since these also influence the hydrology in Germany."